# Multimedia: multimodal mediation analysis of microbiome data

Hanying Jiang,[1] Xinran Miao,[1] Margaret W. Thairu,[2] Mara Beebe,[2] Dan W. Grupe,[3] Richard J. Davidson,[3,4,5] Jo Handelsman,[2,6] Kris Sankaran[1,2]

**ABSTRACT** Mediation analysis has emerged as a versatile tool for answering mechanistic questions in microbiome research because it provides a statistical framework for attributing treatment effects to alternative causal pathways. Using a series of linked regressions, this analysis quantifies how complementary data relate to one another and respond to treatments. Despite these advances, existing software's rigid assumptions often result in users viewing mediation analysis as a black box. We designed the multimedia R package to make advanced mediation analysis techniques accessible, ensuring that statistical components are interpretable and adaptable. The package provides a uniform interface to direct and indirect effect estimation, synthetic null hypothesis testing, bootstrap confidence interval construction, and sensitivity analysis, enabling experimentation with various mediator and outcome models while maintaining a simple overall workflow. The software includes modules for regularized linear, compositional, random forest, hierarchical, and hurdle modeling, making it well-suited to microbiome data. We illustrate the package through two case studies. The first re-analyzes a study of the microbiome and metabolome of Inflammatory Bowel Disease patients, uncovering potential mechanistic interactions between the microbiome and disease-associated metabolites, not found in the original study. The second analyzes new data about the influence of mindfulness practice on the microbiome. The mediation analysis highlights shifts in taxa previously associated with depression that cannot be explained indirectly by diet or sleep behaviors alone. A gallery of examples and further documentation can be found at https://go.wisc.edu/830110.

**IMPORTANCE** Microbiome studies routinely gather complementary data to capture different aspects of a microbiome's response to a change, such as the introduction of a therapeutic. Mediation analysis clarifies the extent to which responses occur sequentially via mediators, thereby supporting causal, rather than purely descriptive, interpretation. Multimedia is a modular R package with close ties to the wider microbiome software ecosystem that makes statistically rigorous, flexible mediation analysis easily accessible, setting the stage for precise and causally informed microbiome engineering.

**KEYWORDS** human microbiome, computational biology, statistics, biostatistics

Treatments often cause change indirectly, triggering a chain of effects that eventually influences outcomes of interest. A standard approach to disentangling these pathways is to distinguish between indirect paths through candidate mediators and direct paths from treatment to outcome. Figure 1A represents this graphically, with separate paths for treatment $T \rightarrow$ mediator $M \rightarrow$ outcome $Y$ and treatment $T \rightarrow$ outcome $Y$. In the causal inference literature, this exercise is called mediation analysis, and various techniques have emerged to support it (1, 2). Several adaptations have been proposed for the microbiome setting, where mediators, outcomes, and controls may be high-dimensional (3–6). These efforts have already uncovered clinically relevant

Address correspondence to Kris Sankaran, ksankaran@wisc.edu.

Hanying Jiang and Xinran Miao contributed equally to this article. Author order was determined alphabetically.

The authors declare no conflict of interest.

See the funding table on p. 17.

relationships, like the existence of microbial taxa that mediate the success of chemotherapy treatments (7).

Despite these successes, existing methodology places strong requirements on the distribution of the mediators or outcome variables and the functional form of their relationships. For example (5, 6, 8, 9), assume that mediators are compositional and that outcomes are univariate, focusing on how microbiome relative abundance profiles mediate treatment effects on downstream host phenotypes, like the relationship between fat intake and body mass index (5). This precludes analysis where outcomes are multidimensional, like metabolic profiles, or where mediators are clinical measurements. Furthermore, with the exception of the mediation package (10), existing implementations are not modular, fixing the estimator used in both the mediator and outcome regressions. This rigidity limits the range of settings in which mediation analysis can be applied. Moreover, it discourages critical evaluation or interactive model building since model components are difficult (or impossible) to interchange. Unfortunately, even the adaptable mediation package is limited to one-dimensional mediator and outcome variables.

To enable more flexible and transparent mediation analysis of microbiome data, we extend the methodology introduced by (10, 11) to high-dimensional mediator and outcome variables. This makes it possible to include sparse regression, logistic-normal multinomial, random forest, hierarchical Bayesian, and hurdle mediator and outcome models within a uniform package interface. Moreover, we have documented the process of inserting custom models into the overall workflow. These models can all be specified using R's formula notation, and components can be easily interchanged according to context. We include operations for summarization, alteration, and uncertainty quantification for the resulting models, encouraging interactive and critical microbiome mediation analysis. We ensure strong ties to the wider microbiome software ecosystem by including methods to convert to and from phyloseq (12) and SummarizedExperiment (13, 14) data structures. Briefly, this research makes the following contributions:

- We define a flexible implementation of the generalized mediation analysis framework that applies to multivariate mediators and outcomes, and we develop modules for nonlinear (random forest), high-dimensional (regularized linear model), zero-inflated (hurdle model), and compositional (logistic-normal multinomial) mediator and outcome models.
- We define a transparent interface linking widely used microbiome data structures to mediation analysis routines, including direct and indirect effect estimation, bootstrap inference, synthetic null hypothesis testing, sensitivity analysis, and summary visualization.
- We provide detailed case studies of how causal mediation analysis can guide principled data integration in multi-omics settings.

Altogether, the multimedia package unlocks the potential for mediation analysis for microbiome studies with complex experimental designs, enabling model-based integration of diverse data types, including microbial community composition, high-throughput molecular profiles, and host health surveys.

## RESULTS

Mediation analysis with our package is a three-step process. First, users specify the hypothesized causal relationships between variables with a concise syntax that represents diverse modeling choices (**Model Setup**). Next, they estimate the model parameters and the associated causal effects (**Counterfactual Analysis**). Finally, they can compare synthetic data from alternative models and calibrate inferences using either bootstrap confidence intervals or hypothesis tests (**Evaluating Uncertainty**). This overall workflow is illustrated in Fig. 1B and detailed in the first three sections below. A summary of key package functions is given in Table 1. The last two sections demonstrate the

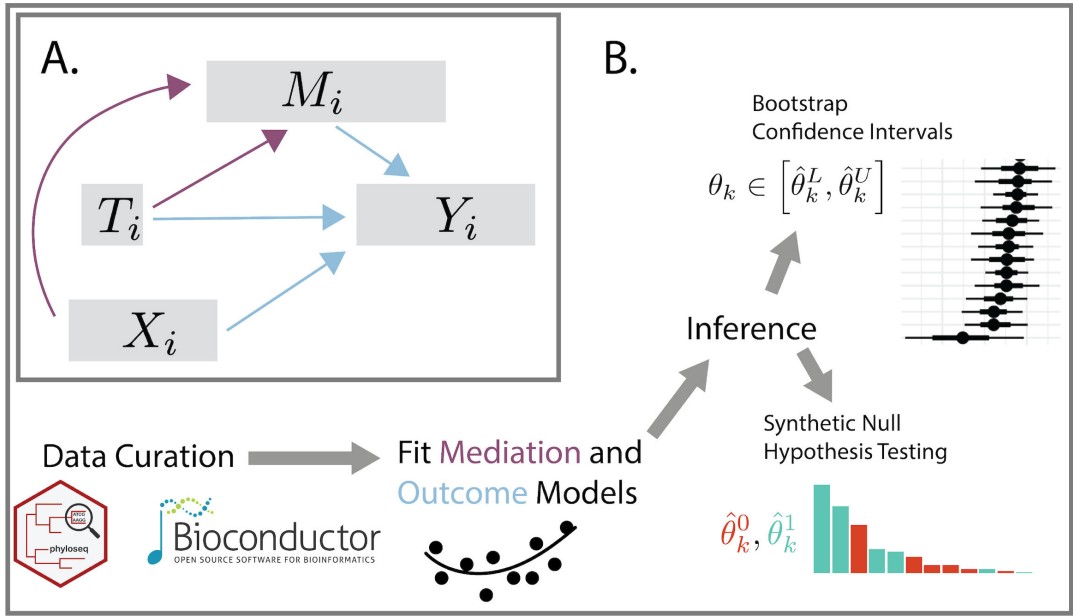

**FIG 1** (A) The graphical model underlying mediation analysis. Using combined mediation (purple) and outcome (blue) models, mediation analysis makes it possible to distinguish between direct and indirect causal pathways between treatments and outcomes. The conventional mediation analysis typically requires all nodes except for the covariates $X$ to be univariate, whereas our package operates without such constraints. (B) The overall multimedia workflow. Multimedia defines a modular interface to mediation analysis with utilities for summarizing and evaluating uncertainty in estimated effects.

package workflow with case studies on metabolomic data integration and the gut-brain axis.

## Model setup

To estimate a mediation model, it is necessary to fully specify the nodes and edges in Fig. 1A. The nodes are used to divide data sources into categories according to their role in the causal model. Edges correspond to mediator and outcome models. Rather than requiring the specification of all mediation analysis components at once in a single function, multimedia allows users to define separate components and then glue them together to define an overall analysis. The package exports a mediation_data data structure for storing the samples used in model fitting. We use R's S4 system (15) to define separate slots for each node in Fig. 1A. This data structure can be created by applying the accompanying mediation_data function to accompanying R data.frame, phyloseq, and SummarizedExperiment objects. We support tidyverse-style syntax (16), meaning that many variables can be assigned to a node using concise queries. For example, mediation = starts_with("diet") will search the input data for any features starting with the string "diet" and will tag them as mediators in the downstream analysis. This efficient matching simplifies data manipulation in high-dimensional settings, where the user may need to work with hundreds of mediators or outcomes.

Next, we must specify the mediator and outcome models. The package exports wrappers to several regression families, ensuring that, despite their differing underlying methodology, all families can be used interchangeably for estimation, sampling, and prediction in the overall mediation analysis workflow. Specifically, multimedia includes (i) linear regression, which ensures that the package generalizes the earlier mediation package, (ii) $\ell^1$ and $\ell^2$-regularized linear regression (17, 18), which can be more stable and interpretable in the presence of numerous predictors, (iii) random forests (19), which supports detection of nonlinear relationships between variables, and (iv) hierarchical Bayesian regression (20), which can be useful for sharing information across related groups. Among the hierarchical Bayesian models, we highlight the available hurdle

**TABLE 1** Core functions for problem specification, effect estimation, and uncertainty quantification available through the multimedia package[a].

| Stage | Function | Description |
|---|---|---|
| Model Setup | mediation_data | Convert phyloseq, SummarizedExperiment, or data.frame objects into S4 classes representing all components of a mediation analysis study |
| | multimedia | Define the form of the mediator and outcome models for estimation and effect calculations |
| Counterfactual Analysis | direct_effect | Estimate direct effects for each outcome (equation (8)) using the estimator in equation (16) |
| | indirect_overall | Estimate overall indirect effects for each outcome (equation (7)) using the estimator in equation (15) |
| | indirect_pathwise | Estimate indirect effects for each mediator-outcome pair (equation (9)) using the estimator in equation (17) |
| Statistical Inference | bootstrap | Re-estimate models and effects on bootstrap resampled versions of the experiment |
| | nullify | Define a version of an existing model with a subset of edges removed from either the mediation or outcome model |
| | fdr_summary | Calibrate a false discovery rate controlling selection rule using synthetic null data and equation (18) |
| Sensitivity Analysis | sensitivity | Evaluate the sensitivity of estimated overall indirect effects to violations of assumption following equation (20) |
| | sensitivity_pathwise | Evaluate the sensitivity of estimated pathwise indirect effects to violations of assumptions following equation (20) |
| | sensitivity_perturb | Evaluate the sensitivity of estimated overall indirect effects to violations of assumptions following equation (21) |

[a]The complete function reference can be read online at https://go.wisc.edu/830110. or as a PDF manual at https://go.wisc.edu/olm213.

regression models, which have previously proven useful for modeling zero-inflated microbiome data (21, 22).

## Counterfactual analysis

After using the estimate function to fit models to the observed data, we can reason about potential outcomes under different treatment regimes. This allows us to clarify the relative importance of direct and indirect pathways. For example, to estimate a direct effect ($T \to Y$), we can block effects that travel along the indirect path ($T \to M \to Y$) and measure the changes to the responses that persist. Formally, in the counterfactual language of the Materials and Methods, direct and indirect effects are estimated using predicted mediators $\widehat{M}(t)$ and outcomes $\widehat{Y}(t', \widehat{M}(t))$, where $t$ and $t'$ correspond to mediator and outcome-specific treatment assignments. To this end, multimedia defines a data structure for storing $(t, t')$ within two data.frames whose rows are samples and columns are treatment settings. The predict and sample methods allow users to compute expected values and draw samples according to arbitrary treatment profiles $(t, t')$. Note that, in addition to the standard treatment vs control setup, multimedia supports treatment profiles with multiple concurrent treatments and multilevel or continuous treatment.

Given a fitted model, multimedia outputs estimated direct and indirect effects. We formally define these effects in equations (7) to (9). Here, we offer an overview of their motivation and interpretation. Direct effects are the changes we would observe in the outcome if we changed the treatment node in Fig. 1A but held all the mediators fixed. This is the effect that travels along the edge $T \to Y$, and it measures the extent to which the treatment can influence the outcome while bypassing the mediators. We evaluate different direct effects for each outcome. For example, in the mindfulness case study below, direct effects can be interpreted as microbiome shifts (changes in $Y$) following the mindfulness training (treatment $T$) that are not a consequence of changes in participant sleep or diet behaviors (mediators $M$). Next, we support the estimation of two types of indirect effects. Total indirect effects measure the changes in the outcome when setting all mediators to their potential values when the treatment is present, keeping the contribution of the direct path $T \to Y$ fixed. This aggregates the effect across the full collection of indirect paths. In contrast, pathwise indirect effects measure the changes in outcome when comparing counterfactuals that are equal except at a

single mediator. This isolates the indirect effect along a single indirect path. In this case, an indirect effect is reported for each outcome-mediator pair, rather than only for each outcome. Note that the definitions of these effects involve unobservable quantities. Their identification relies on assumptions about the absence of confounding both before and after treatment assignment across configurations of mediators and outcomes, which are detailed in the Section "Counterfactual framework" in the Materials and Methods.

To increase modeling transparency, multimedia includes functions for interacting with and altering fitted models. Direct and indirect effects can be visualized within the context of the original data. This can serve as a sanity check and guide further model refinements. Outputs are created with ggplot2 (23), which allows users to customize plot appearance. The case studies include outputs from these helper visualization functions. Furthermore, given a fitted model, we allow users to refit new versions with sets of edges removed. Figure 2 illustrates the main idea with a toy data set. In the second column, the mediator takes on a larger value under the red treatment, while in the third, the mediators have identical distributions under the two treatments. Similarly, in the fourth, the relationship between the mediator and outcome no longer depends on treatment status. We can also alter the overall model structure, like the switch to a linear outcome model in the last column. If the model quality deteriorates significantly in an altered submodel, then those edges play a critical role. This heuristic is formalized in the synthetic null hypothesis testing strategy discussed below. Finally, we have built the package with extensibility in mind. If functions can be written for estimation and prediction from a new model type, then it can be passed in to multimedia as a custom mediation or outcome model.

## Statistical inference

The multimedia package offers bootstrap (24–26) and synthetic null hypothesis testing (27–29) approaches for quantifying uncertainty in estimates of mediation effects. To bootstrap in the mediation analysis context, we refit the mediator and outcome models to bootstrap resampled versions of the data and compute summary statistics (e.g., direct effect estimates) on each bootstrap sample. The percentiles of the resulting summary statistic distribution define the bootstrap confidence interval. Importantly, the bootstrap is model agnostic and can apply to any instantiation of the counterfactual mediation analysis framework. The primary assumption made by the bootstrap is that its test statistics vary smoothly to small perturbations of the data. For this reason, it is worthwhile to check that the histogram associated with the full bootstrap distribution is well-behaved before computing confidence intervals. Like the boot function in base R,

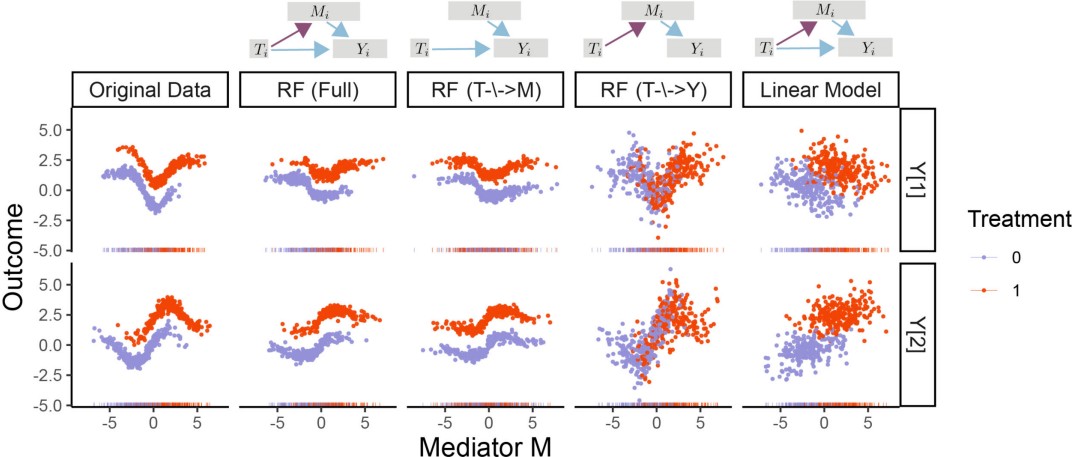

**FIG 2** Samples from altered versions of a mediation analysis model fitted to the toy data at the far left. Each row describes a different outcome variable, and colors represent different treatments. The first column gives the original data, and the remaining columns give simulated data from alternative models specified by the DAGs on the top and column titles.

multimedia's bootstrap uses a functional implementation—any function that transforms an experiment and fitted model into a summary statistic can be used as input. For example, it can accept a list of direct and indirect effect estimators, and these will be computed on bootstrap resample.

An alternative approach to inference in high dimensions is based on synthetic null hypothesis testing. In this approach, rather than resampling the original data, the modeler simulates synthetic data from an assumed null distribution. Effect estimates are computed using both the original and the synthetic null data, and the fraction of synthetic null "negative controls" among the strongest observed effects can be used to calibrate a selection rule with false discovery rate control. The alteration functions above can be used to define synthetic nulls; e.g., after zeroing out the edges from either $T \rightarrow M$ or $M \rightarrow Y$, any estimated indirect effects can be treated as negative controls. Two advantages of the synthetic null approach are that (i) it only requires the mediator and outcome models be estimated twice and (ii) multiple hypothesis testing is accounted for via the false discovery rate. The key disadvantage of this approach, relative to the bootstrap, is that it requires a realistic synthetic null data-generating mechanism. For example, if the synthetic null data are generated from a linear model, but real effects are nonlinear, then the resulting selection sets will not provide valid false discovery rate control.

## Microbiome-Metabolome integration

We next illustrate the multimedia workflow with case studies. Our first concerns inflammatory bowel disease (IBD), which is closely tied to gut microbiome community composition. The studies (30, 31) investigated the relationship between the gut microbiome and metabolome between IBD patients and healthy controls, concluding that microbial community members may be partly responsible for the formation of metabolites that lead to inflammation and IBD. By applying clustering and canonical correlation analysis to untargeted mass spectrometry data, they flagged a number of disease-relevant metabolites. We re-analyze the data using model-based mediation analysis, viewing IBD status—healthy control, ulcerative colitis (UC), or Crohn's disease (CD)—as treatments $T$, metabolic profile as the outcome $Y$, and microbiome community composition as a mediator $M$. The data are downloaded from the microbiome-metabolome curated data repository (32). We have further filtered to the top 173 and 155 most abundant microbes and metabolites, and we apply centered log-ratio (CLR) and $\log(1 + x)$ transformations to each source, respectively. Further details about the experimental cohort and data preparation are available in the Materials and Methods.

We use parallel linear and $\ell^1$-regularized regression for mediator and outcome models, respectively. Note that treatment is the only predictor in the mediator model, which is why no regularization is required. We ran the bootstrap for 1,000 iterations, and 95% confidence intervals and bootstrap distributions for the features with the strongest direct and overall indirect effects contrasting CD with healthy controls are shown in Fig. 3. Metabolites with strong indirect effects are influenced by IBD-induced changes in microbiome community composition, while those with large direct effects change due to other unknown factors. Figure 4 explores a small subset of these overall effects by overlaying metabolite abundances onto multidimensional scaling (MDS) plots derived from microbiome community profiles. Though metabolites with strong direct effects have differential abundance across IBD and healthy groups, only metabolites with indirect effects show variation that is also associated with microbiome composition. We caution that these results are potentially conservative. To ensure stability in high dimensions, the $\ell^1$ and $\ell^2$-regularized regression estimators implemented in multimedia are biased towards 0 (33). This may cause both direct and indirect effects to appear inappropriately weak, and extensions to debiased alternatives like (34) are an important line of future work.

Moreover, by analyzing pathwise indirect effects, we can uncover genus-level relationships. A subset of the strongest pathwise indirect effects are shown in Fig. 5.

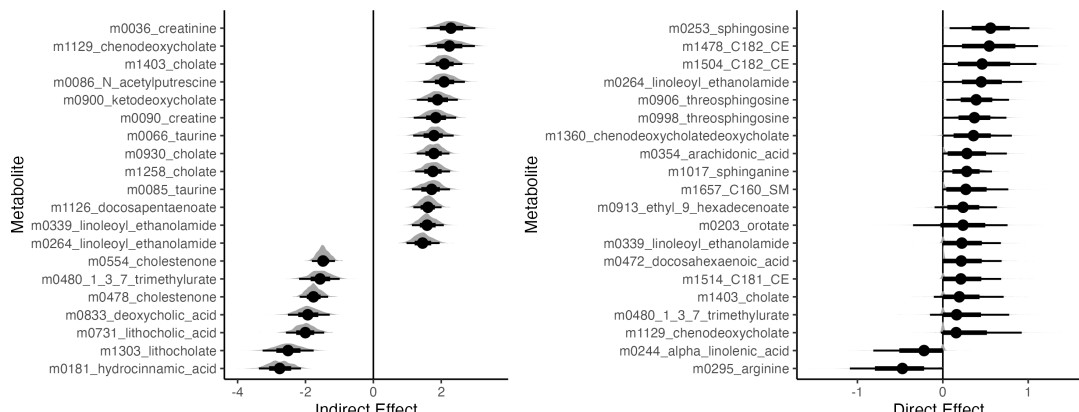

**FIG 3** 95% Bootstrap confidence intervals for metabolites with the strongest estimated direct and overall indirect effects associated with CD. Effects are sorted according to magnitude, and only the top 15 of each type are shown. Within the interval, the inner rectangle captures 66% of the bootstrap samples. In this data, indirect effects are stronger than direct effects.

Among the microbe-metabolite pairs with the strongest pathwise indirect effects, we find a relationship between the metabolite taurine and genus *Bilophila* (Fig. 5). High levels of fecal taurine, one of the primary conjugates of primary bile acids (35), have been previously associated with IBD (36, 37). It has also been found that *Bilophila wadsworthia*, one of the most prominent taurine metabolizers, is often associated with lower levels of taurine (37). Here, our results suggest that higher levels of taurine in IBD patients are mediated, in part, by the abundance of *Bilophila*. We also find microbes in the genus *Firmicutes* bacterium CAG:103 and are paired with several metabolites: cholate, chenodeoxycholate, and 7-ketodeoycholate (Fig. 5). Cholate and chenodeoxycholate are primary bile acids produced by the host, which are the metabolized by gut bacteria to form secondary bile acids. 7α-Dehydroxylation is one of the pathways that bacteria metabolize primary bile acids, an intermediate of which is 7-ketodeoycholate (38). Recent work has found that bacteria closely related to *Firmicutes* bacterium CAG:103

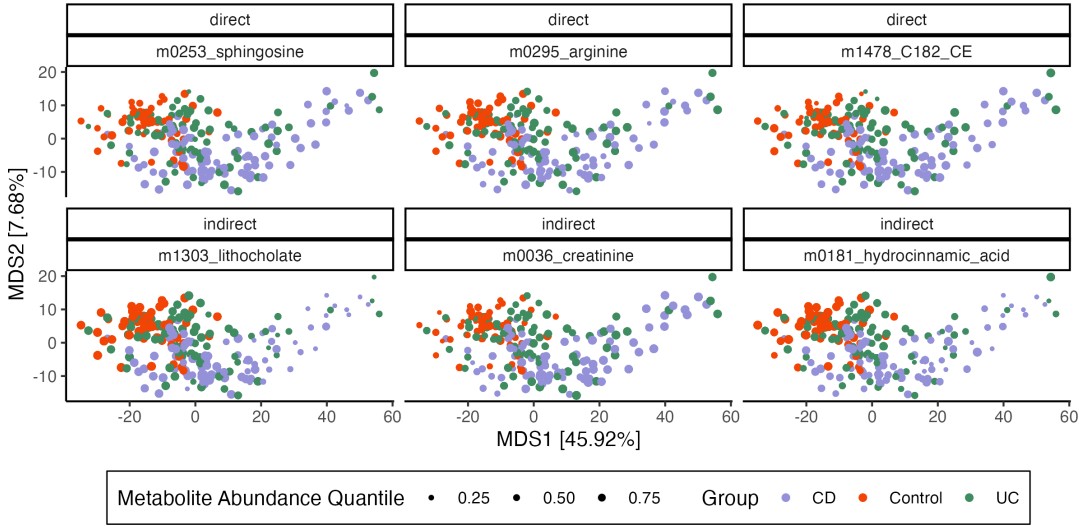

**FIG 4** Microbiome composition and metabolite abundance for three metabolites with the strongest direct (top row) and indirect (bottom row) effects. Samples (points) are arranged according to an MDS on CLR-transformed microbiome profiles with Euclidean Distance. Axis titles give $\frac{\lambda_k}{\sum_{k'} \lambda_{k'}}$ from the associated eigenvalues. Each panel corresponds to a metabolite, and point size encodes metabolite abundance, normalized to panel-specific quantiles. Metabolites with strong indirect effects vary more systematically with microbiome composition—for example, samples with a low abundance of lithocholate are localized to the right of the MDS plot.

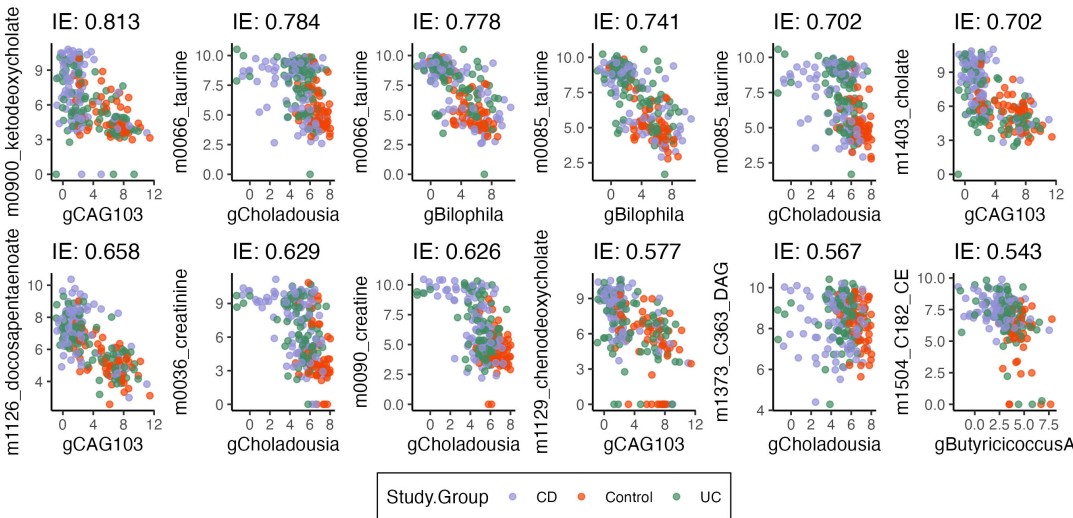

**FIG 5** Microbe-metabolite pairs with the strongest pathwise indirect effects from IBD status. Each panel corresponds to one pair, CLR-transformed genus abundance is given on the $x$-axis, and $\log(1 + x)$-transformed metabolite abundance is given on the $y$-axis. Effects are sorted from most negative (top left) to most positive (bottom right). For a pathwise indirect effect to be strong, there must be both a shift in microbe abundance due to IBD state ($T \rightarrow M$) and also an association between microbe and metabolite abundance ($M \rightarrow Y$).

contain the majority of predicted genes associated with the 7$\alpha$-dehydroxylation pathway within metagenomic samples (39). Our results suggest that the increasing abundance of *Firmicutes* bacterium CAG:103 may be driving the decrease in these primary bile acid metabolites and intermediates, which is associated more with the non-IBD controls (40). Host deficiency in creatine uptake has been associated with poor mucosal health in IBD patients (41). In our results, we find that there is a strong microbe-metabolite pair between microbes in the genus *Choladousia* (family: *Lachnospiraceae*) and creatine/creatinine levels. *Lachnospiraceae*, which is often at lower levels in IBD patients, are known to produce short-chain fatty acids that have been shown to help with mucosal health (42) (Fig. 5). Overall, these results suggest that *Choladousia* may utilize creatine/creatinine as a nitrogen source, thus explaining its higher abundance in the controls.

Our discussion assumed no unmeasured confounding between mediators and outcomes. Sensitivity analysis can clarify whether these conclusions remain true even when assumptions are violated. Using the approach detailed in the Materials and Methods (equation (19)), we assessed pathwise indirect effects for three metabolite-genus pairs. The results in Fig. 6 show the robustness of the taurine-Bilophila and sensitivity of the taurine-Choladousia indirect effect estimates. The ketodeoxycholate-CAG103 effect is intermediate between these extremes, with indirect effects present up to confounding strength $\rho = 0.5$. More generally, multimedia offers functionality for evaluating sensitivity for a range of user-specified pretreatment confounding patterns. Our online vignette provides an additional example of sensitivity analysis for total, rather than pathwise, indirect effects.

Note that, since this mediation model is built from a regularized linear regression outcome model, it is more sensitive to linear associations between microbe and metabolite abundances. The official package documentation includes an alternative Bayesian hurdle outcome model, which exhibits higher sensitivity to outcomes with changes in metabolite presence-absence probability. The easy interchangeability of mediation analysis components makes this contrasting analysis simple to implement—it only requires change in a single line of code—and reflects multimedia's modular design.

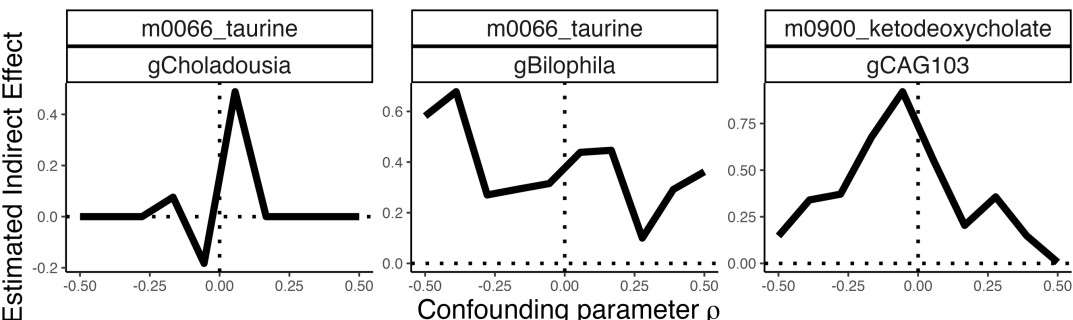

**FIG 6** Sensitivity analysis for three metabolite-genus pairs in the IBD study. The strength of unmeasured confounding between mediators and outcomes is reflected in the $x$-axis parameter $\rho$. When the sign of the estimated indirect effect flips for small values of $|\rho|$, then the estimate is sensitive to violations in the identification assumptions.

## Evaluating a mindfulness intervention

Studies of the gut-brain axis have yielded experimental evidence for interactions between the gut microbiome and the brain. For example, germ-free mice colonized with the microbiota from human patients with clinical depression develop depression-like symptoms (43, 44), and observational studies have linked particular bacterial taxa to depression (45, 46). Given this growing body of evidence, a team from the UW-Madison Center for Healthy Minds and the Wisconsin Institute for Discovery profiled microbiome composition, surveyed psychological symptoms, and tracked behavior change among 54 subjects before and after participation in a 2-month mindfulness training (47, 48)—see the Methods and Materials for details of the study design and data processing. This study aimed to determine the nature of the mindfulness-microbiome relationship and to identify potential causal pathways. Such understanding could lead to novel interventions that influence mood through the microbiome. As a first step, we use mediation analysis to understand the mechanisms linking mindfulness and the microbiome in this randomized controlled trial. Our intervention $T$ is the mindfulness training program, the outcome of interest is microbiome composition $Y$, and mediators $M$ are survey responses related to diet and sleep that are hypothesized to influence the microbiome. To control for subject-to-subject level variation, participant ID is used as a pretreatment variable $X$.

For mediator and outcome models, we apply ridge and logistic-normal multinomial regressions, respectively (49, 50). We choose a ridge regression model so that intercepts across the large number of participants are shrunk toward their global mean. We choose logistic-normal multinomial regression to jointly model microbiome composition. We also define altered submodels where all direct and indirect effects have been removed. Simulated genera compositions from all models are shown in Fig. 7. In the newly simulated data, subjects have been randomly re-assigned to the treatment and control groups. These submodels can support synthetic null hypothesis testing since the synthetic null data appear to capture relevant properties of the real microbiome composition profiles, like the average relative abundances across genera and the range of observed abundances within most genera. Their main limitation is that some genera, like *Methanobrevibacter*, *Paraprevotella*, and *Akkermansia*, have much wider ranges than the synthetic data, and Fig. S1 suggests that this is due to a failure to capture the unusually high zero inflation present in these genera.

For synthetic null hypothesis testing, models without $T \rightarrow Y$ and $M \rightarrow Y$ associations are used to generate negative controls for direct and total indirect effect estimates, respectively. Figure 8 shows the estimated effects from real and synthetic data, together with the estimated false discovery rates. At a level $q = 0.15$, five genera are selected as having either significant direct or indirect effects. Figure S1 provides the analog of Fig. 5 for this case study. Indirect effects are an order of magnitude weaker than direct

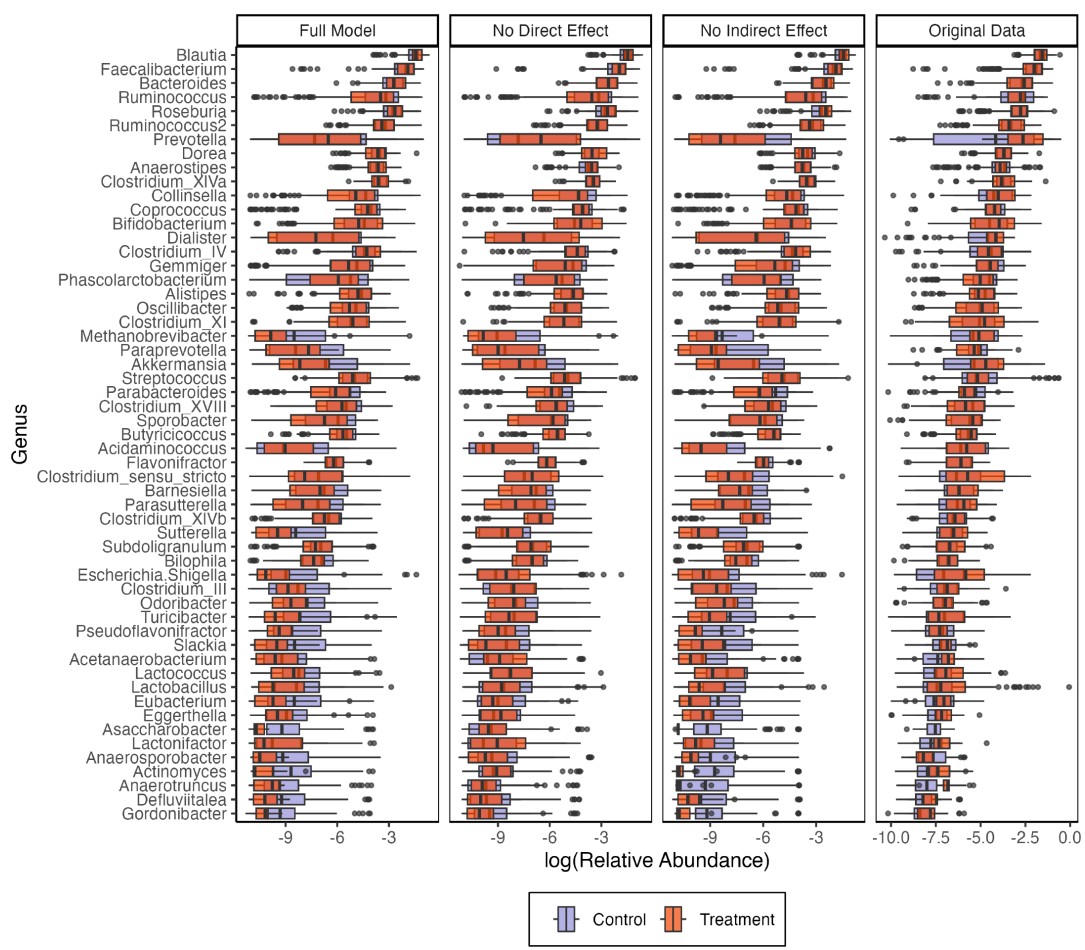

**FIG 7** Real and synthetic null relative abundances across a subset of genera at different overall relative abundances. Color distinguishes whether the participant belonged to the treatment (mindfulness training) or control groups. The full model (left panel) captures the overall abundances and trajectories present in the real data though it tends to underestimate the heaviness of the tails. The second and third panels show the analogous models with direct ($T \to Y$) and indirect ($M \to Y$) effects removed.

effects, suggesting that changes in microbiome composition following the mindfulness intervention cannot simply be attributed to changes in diet or sleep alone.

We cannot externally validate these findings since there is no consensus on the relationship between specific taxonomic groups and common psychiatric disorders [for a description of current sources of controversy, see reference (51)]. However, our findings are broadly consistent with those from a recent large-scale human cohort, which found that most genera belonging to the families *Ruminococcaceae* were depleted in people with more symptoms of depression and that *Bifidobacterium* was an important predictor of depressive symptoms in a random forest classifier (45).

## DISCUSSION

Mediation analysis makes it possible to study causal pathways in multimodal microbiome data, and it is an essential tool for discovery of subtle relationships that span multiple host measurements and high-throughput assays. Statistical techniques in this space are needed to support the interrogation of varied causal relationships, not simply studies where microbiome profiles serve as mediators and outcomes are one-dimensional, as has been the historical focus of the field.

Our case studies illustrate the flexibility and analytical depth supported by multimedia. Unlike traditional microbiome mediation analysis software, the package allows the specification of diverse regression components, and the interface simplifies the

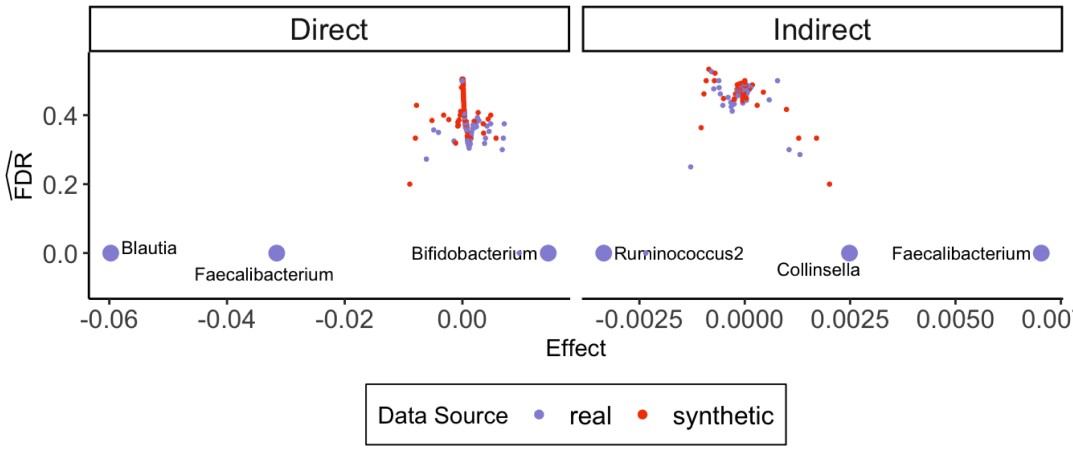

**FIG 8** Estimated direct and total indirect effects and false discovery rates derived from real and synthetic null data. Each point corresponds to one genus in either real (blue) or simulated (orange) data. The genera selected to control the false discovery rate at $q \leq 0.15$ are drawn larger than the rest. Direct effects are both larger in magnitude and easier to distinguish than their indirect counterparts.

interpretation of effect types and model criticism. In this way, multimedia encourages interactive, rigorous mediation analysis for microbiome data. It is written to interface closely with the existing microbiome software ecosystem, and since analyses are carried out in reproducible code notebooks, it supports scientific transparency.

We note that multimedia is related to other recent approaches to transparent microbiome mediation analysis, most notably MiMed (52), which provides a self-contained graphical interface to support this task. The MiMed interface is available as a web server and a standalone Shiny App (53). MiMed and multimedia make recent statistical advances in microbiome mediation analysis more accessible and offer advanced customizability. Furthermore, both software packages implement the generalized causal mediation analysis framework (11); the effect estimates and confidence intervals output by the packages share the same conceptual foundation. Nonetheless, there are critical distinctions. For example, MiMed is accessible to users with no programming experience, while multimedia requires familiarity with R software. Limiting multimedia to those with programming experience allows for a more modular design, with easily interchangeable and extensible code components. In particular, multimedia offers a more thorough instantiation of the generalized mediation analysis framework. MiMed's implementation requires linear mediator and outcome models, and the outcome models must have univariate responses. In contrast, multimedia offers a broader range of model types (e.g., regularized linear or logistic-normal multinomial) that fit within the framework of reference (11), and both mediator and outcome models can be multivariate. As seen in both case studies, this additional flexibility enables the integration of more complex multivariate mediator and outcome data.

We have created a gallery of example notebooks that use the multimedia package. These include alternative analyses of the IBD and mindfulness data explored here. We invite users to contribute further examples, and we plan to structure further developments according to community needs.

## MATERIALS AND METHODS

### Counterfactual framework

Let $T \in \mathcal{T}$ be the treatment, $M \in \mathcal{M}$ be the mediators of interest, $Y \in \mathcal{Y}$ be the outcome, and $X \in \mathcal{X}$ be the pretreatment covariates, where $\mathcal{T} \subset \mathbb{R}, \mathcal{M} \subset \mathbb{R}^K, \mathcal{Y} \subset \mathbb{R}^J$, and $\mathcal{X} \subset \mathbb{R}^P$ represent the supports of $T, M, Y$, and $X$. For simplicity, we assume

$\mathcal{T} = \{0, 1\}$ and $T$ is a binary indicator of either treatment ($T = 1$) or control ($T = 0$), though multimedia supports categorical, continuous, and multi-treatment cases.

We first consider the total indirect effect through all mediators and the direct effect through other mechanisms. Applying a counterfactual perspective, we define $M(t)$ as the potential values of the mediators under $T = t$ and $Y(t, m)$ as the potential outcome under $T = t$ and $M = m$. Therefore, we can use $Y(t, M(t'))$ to denote the potential outcome under the treatment status $t$ when the mediators are set to be the potential values under $t'$. In reality, we can only ever observe the case where $t$ and $t'$ are the same, i.e., $Y(1, M(1))$ in the treated group and $Y(0, M(0))$ in the control group—but conceptually $t$ and $t'$ can be different. For example, $Y(0, M(1))$ represents the potential outcome when only the mediators are intervened upon and $Y(1, M(0))$ represents the potential outcome when we make interventions while keeping the mediators at their values under the control. For notational simplicity, we omit the dependence of $M$ and $Y$ on $X$.

We adopt the definitions in reference (11), where the indirect effect is defined as

$$\delta(t) = \mathbb{E}\{Y(t, M(1)) - Y(t, M(0))\} \tag{1}$$

and the direct effect is defined as

$$\zeta(t') = \mathbb{E}\{Y(1, M(t')) - Y(0, M(t'))\} \tag{2}$$

for $t, t' \in \{0, 1\}$. It has been shown in reference (54) that both effects are nonparametrically identifiable under the sequential ignorability assumption:

$$\{Y(t', m), M(t)\} \perp\!\!\!\perp T \mid X = x, \tag{3}$$
$$Y(t', m) \perp\!\!\!\perp M(t) \mid T = t, X = x, \tag{4}$$
$$\mathbb{P}(T = t \mid X = x) > 0, \tag{5}$$
$$p_{M(t)}(m \mid T = t, X = x) > 0, \tag{6}$$

for any $t, t', m, x$.

Without additional assumptions, $\delta(t)$ and $\zeta(t)$ may vary with $t$. To provide a consistent and interpretable summary, we measure the total indirect effect and direct effect defined as follows;

$$\bar{\delta} = \frac{1}{2} \sum_{t=0}^{1} \mathbb{E}\{Y(t, M(1)) - Y(t, M(0))\} \tag{7}$$

$$\bar{\zeta} = \frac{1}{2} \sum_{t'=0}^{1} \mathbb{E}\{Y(1, M(t')) - Y(0, M(t'))\} \tag{8}$$

Large magnitudes of $\bar{\delta}$ and $\bar{\zeta}$ suggest strong indirect and direct effects.

Moreover, we can also examine the pathwise indirect effect through each mediator. We assume there is no causal relationship between the mediators $M = (M_1, ..., M_K)$. When interest lies in the mediator $M_k$, we emphasize the dependence of the potential outcome on both $M_k$ and the remaining mediators $M_{-k}$ by writing $Y(t, m, w)$, explicitly distinguishing $M_k = m$ and $M_{-k} = w$. To evaluate the pathwise indirect effect through $M_k$, we consider different treatment assignments for $M_k$ and $M_{-k}$. For example, $Y(t, M_k(t'), M_{-k}(t''))$ represents the potential outcome under the treatment status $t$ when $M_k$ is set to be its potential value under $t'$ and $M_{-k}(t'')$ are set to be their potential values under $t''$. Using these notations, we can define the pathwise indirect effect through $M_k$ as:

$$\bar{\omega}_k = \frac{1}{2} \sum_{t'=0}^{1} \mathbb{E}\{Y(t', M_k(1), M_{-k}(t')) - Y(t', M_k(0), M_{-k}(t'))\} \tag{9}$$

This quantity has been proven to be nonparametrically identifiable under a generalized version of the sequential ignorability assumption (55):

$$\{Y(t,m,w), M_k(t'), M_{-k}(t'')\} \perp\!\!\!\perp T \mid X = x \tag{10}$$
$$Y(t', m, M_{-k}(t')) \perp\!\!\!\perp M_k \mid T = t, X = x, \tag{11}$$
$$Y(t', M_k(t'), w) \perp\!\!\!\perp M_{-k} \mid T = t, X = x, \tag{12}$$
$$\mathbb{P}(T = t \mid X = x) > 0, \tag{13}$$
$$p_{(M_k t, M_{-k}(t))}(m, w \mid T = t, X = x) > 0, \tag{14}$$

for any possible $t, t', t'', m, w, x$.

## Mediator and outcome model definition

Multimedia estimates the population quantities $\bar{\delta}, \bar{\zeta}$, and $\bar{\omega}$ by replacing the expectations in equations (7) to (9) with the average of fitted values under the estimated mediator and outcome models:

$$\widehat{\bar{\delta}} = \frac{1}{2} \sum_{t=0}^{1} \sum_{i=1}^{n} \widehat{Y}_i(t, \widehat{M}_i(1)) - \widehat{Y}_i(t, \widehat{M}_i(0)), \tag{15}$$

$$\widehat{\bar{\zeta}} = \frac{1}{2} \sum_{t'=0}^{1} \sum_{i=1}^{n} \widehat{Y}_i(1, \widehat{M}_i(t')) - \widehat{Y}_i(0, \widehat{M}_i(t')), \tag{16}$$

$$\widehat{\bar{\omega}} = \frac{1}{2} \sum_{t'=0}^{1} \sum_{i=1}^{n} \left\{ \widehat{Y}_i\left(t', \widehat{M}_{ik}(1), \widehat{M}_{i,-k}(t')\right) - \widehat{Y}_i\left(t', \widehat{M}_{ik}(0), \widehat{M}_{i,-k}(t')\right) \right\} \tag{17}$$

A benefit of applying this generalized causal mediation analysis framework is that various prediction models can be used to obtain estimates $\widehat{M}(t,x)$ and $\widehat{Y}(t,m,x)$ of $M(t,x)$ and $Y(t,m,x)$, respectively. This flexibility is especially valuable in the microbiome context, where both $Y$ and $M$ may be multivariate and where observations may be zero-inflated, high-dimensional, compositional, or highly skewed. For example, the mediators and outcomes may represent survey responses, community taxonomic compositions, or metabolomic profiles. The approach of the multimedia package is to define an interface where prediction methods that have been designed to address these complexities can be easily swapped in and out. Therefore, advances in the prediction of microbiome data can be easily incorporated to improve causal effect estimation through higher-quality mediator and outcome models.

Specifically, the estimates in Formula (15)–(17) allow these prediction algorithms to be used as building blocks in support of estimating direct and indirect causal mediation effects. For example, on its own, random forests are only useful for prediction. But through $\widehat{M}(t,x)$ or $\widehat{Y}(t,m,x)$, they can provide plug-in estimates for causal analysis. We next provide details of the specific estimates used in our case studies though we again emphasize the broader generality of the underlying implementation. In the Section "Microbiome-Metabolome Integration," we fit a separate sparse linear regression model to each metabolite with all CLR-transformed microbe abundances as inputs. Letting $Y_{ij}$ represents the peak intensity for metabolite $j$ in sample $i$ and $M_i$ the relative abundances of microbes in sample $i$, we estimate

$$\widehat{\beta}_j := \arg \min_{\beta_j \in \mathbb{R}^K} \sum_{i=1}^{n} \left( \log(1 + Y_{ij}) - \mathrm{CLR}(M_i)^T \beta_j \right)^2 + \lambda \|\beta_j\|_1$$

In this case, the outcome model $\widehat{Y}(t,m,x)$ is a collection of metabolite-specific estimates $\widehat{\beta}_1, \ldots, \widehat{\beta}_J$ fit simultaneously. Note that the regularization parameter $\lambda$ is fixed across all responses, rather than adaptive to metabolite $j$. The package supports linear, elastic net (56), random forest (19), hurdle (57), and hierarchical (including hurdle) models (20) for either mediator $\widehat{M}(t,x)$ or outcome $\widehat{Y}(t,m,x)$ models similarly.

Alternatively, instead of a collection of univariate models, a multivariate regression model can be fit to relate covariates with the high-dimensional response. This is the approach used in the Section "Evaluating a Mindfulness Intervention," where a single logistic-normal multinomial model (50) is applied to model community composition as a function of treatment $T_i$, survey-derived mediators $M_i$, and pretreatment features $X_i$. In this case, the outcome model is a single, multivariate model estimated using the maximum a posteriori parameter $\hat{B}$ from a logistic-normal multinomial model with a normal prior:

$$\hat{B} := \arg \max_{B \in \mathbb{R}^{(J-1) \times (1+K+P)}} \left[ \prod_{i=1}^{N} \mathrm{Mult}\left( \sum_j Y_{ij}, \varphi^{-1}(BZ_i) \right) \right] p(B).$$

$$Z_i := \left[ T_i \mid M_i \mid X_i \right]^{\top}$$

$$p(B) := \prod_{kp} \mathcal{N}\left( b_{kp} \mid 0, \sigma^2 \right)$$

where $\varphi^{-1} : \mathbb{R}^{J-1} \to \mathbb{R}^J$ is the mapping

$$\varphi^{-1}(\mu) = \left( \frac{\exp(\mu_1)}{1 + \sum_j \exp(\mu_j)}, \dots, \frac{\exp(\mu_{J-1})}{1 + \sum_j \exp(\mu_j)}, \frac{1}{1 + \sum_j \exp(\mu_j)} \right)$$

Note that all bootstrap, synthetic null testing, and sensitivity analysis functions are designed to operate on an abstract mediation_model S4 class. In this way, multimedia is easily extensible, and its causal mediation framework can be applied to various models, including those supplied by a user, as long as they satisfy the S4 class requirements.

## Bootstrap and synthetic null testing

Form a bootstrap resample of the data $\mathcal{D}^* = (\mathbf{X}^*, \mathbf{M}^*, \mathbf{T}^*, \mathbf{Y}^*)$ by independently resampling the $n$ observations with replacement. A summary statistic computed on the $b^{th}$ resampled data set is denoted by $\hat{\theta}^{*b}(\mathcal{D}^*)$. For brevity, we will omit the data arguments. For example, $\hat{\theta}^{*b}$ could correspond to an estimator of $\bar{\delta}$ or $\bar{\zeta}$ derived from mediator and outcome models learned from $\mathcal{D}^*$. Repeat this process $B$ times and refit $\widehat{M}(t, x)$, $\widehat{Y}(t, m, x)$ and the provided summary statistic $\hat{\theta}$ for each of the bootstrapped data sets, yielding the bootstrap distribution $\left( \hat{\theta}^{*b} \right)_{b=1}^{B}$. Let $q_{\frac{\alpha}{2}}$ and $q_{1-\frac{\alpha}{2}}$ represent the $\frac{\alpha}{2}$ and $1 - \frac{\alpha}{2}$ quantiles of this set. Then, $\left[ q_{\frac{\alpha}{2}}, q_{1-\frac{\alpha}{2}} \right]$ forms an $\alpha$-level bootstrap confidence interval for $\hat{\theta}$.

For synthetic null hypothesis testing, estimate mediator and outcome models $\widehat{M}_{\mathrm{sub}}(t, x)$, $\widehat{Y}_{\mathrm{sub}}(t, m, x)$ using only a subset of edges within the DAG. This defines the null data generating mechanism. Using the same pretreatment and treatment profiles $X_i, T_i$ from the original experiment, simulate synthetic null data $\mathbf{M}^{*0}, \mathbf{Y}^{*0}$ from the submodel. For $D$ taxa of interest, compute summary statistics $\left( \hat{\theta}_d^1 \right)_{d=1}^{D}$ and $\left( \hat{\theta}_d^0 \right)_{d=1}^{D}$ based on the original and the synthetic null data, respectively. For example, $\hat{\theta}_d^1$ could estimate taxon $d$'s direct effect $\widehat{\bar{\delta}}_d$ using the original data, and $\hat{\theta}_d^0$ could be the corresponding estimate derived from synthetic null data. Next, for any threshold $t$, we estimate the false discovery rate using

$$\widehat{\mathrm{FDR}}(t) := \frac{\#\left\{d: \left|\hat{\theta}_d^0\right| > t\right\}}{\#\left\{d: \left|\hat{\theta}_d^0\right| > t\right\} + \#\left\{d: \left|\hat{\theta}_d^1\right| > t\right\}} \,. \tag{18}$$

The numerator counts the number of estimates from the synthetic null data that are larger than $t$, and the denominator counts the number of discoveries across either simulated or real data at that threshold. Given a desired FDR level $q$, the selection rule is defined by selecting $t^* = \min\left\{t: \widehat{\mathrm{FDR}}(t) \le q\right\}$ and selecting all features $d$ such that $\left|\hat{\theta}_d^1\right| > t^*$. Under the null samples generated by $\widehat{M}_{\mathrm{sub}}(t, x), \widehat{Y}_{\mathrm{sub}}(t, m, x)$, this rule controls the false discovery rate below level $q$, regardless of the choice of estimator $\hat{\theta}_d$, though better estimators lead to improved power.

## Sensitivity analysis

Mediation analysis relies on untestable identification assumptions, detailed in the "Counterfactual framework" section. While these assumptions cannot be directly tested, the consequences of their violation can be explored through sensitivity analysis. We next review the sensitivity analysis methods available in the multimedia package, which are motivated by the more general methodology (54). Sensitivity is evaluated by simulating counterfactual mediator and outcome variables with correlated noise terms, representing the situation where the assumption of no pretreatment confounding is violated. Specifically, we sample:

$$Y^*(t, m) = \widehat{Y}(t, m) + \epsilon^y \quad \text{and} \quad M^*(t) = \widehat{M}(t) + \epsilon^m. \tag{19}$$

where $\mathrm{Cov}(\epsilon^m, \epsilon^y) \ne \mathbf{0}$. Given these data, we re-estimate either the total or pathwise indirect effects. This helps identify cases where the estimated indirect effects become zero or change signs when confounding is present compared to when $\mathrm{Cov}(\epsilon^m, \epsilon^y) = \mathbf{0}$.

Specifically, the package offers tools for simulating and assessing effects under covariance structures for $(\epsilon^m, \epsilon^y)$ that represent pretreatment confounding. For example, users can generate data from equation (19) with:

$$\Sigma(\rho, G) := \begin{pmatrix} \mathrm{diag}(\hat{\sigma}_M^2) & \rho \hat{\sigma}_M \hat{\sigma}_Y^\top \odot \mathbf{1}_G \\ \rho \hat{\sigma}_Y \hat{\sigma}_M^\top \odot \mathbf{1}_G^\top & \mathrm{diag}(\hat{\sigma}_Y^2) \end{pmatrix} \tag{20}$$

$\hat{\sigma}_M^2 \in \mathbb{R}_+^K$ and $\hat{\sigma}_Y^2 \in \mathbb{R}_+^J$ represent the estimated noise variances of mediators and outcomes, and $\mathbf{1}_G \in \{0, 1\}^{K \times J}$ is an indicator over mediator-outcome pairs $G$ on which to evaluate sensitivity. When $\rho \ne 0$, unmeasured confounding is present between these pairs. We recommend keeping $G$ small because confounding patterns induced by large $G$ are less plausible. For example, it is unlikely that a single mediator can be confounded with all outcomes, while all other mediators remain unconfounded. By adjusting $\rho$ and $G$, package users can evaluate sensitivity to various patterns of pretreatment confounding.

The package also offers a more general form of sensitivity analysis, where users can supply an arbitrary matrix $\Delta$ and simulate noise from:

$$\Sigma(\Delta, \nu) = \mathrm{diag}\left(\left[\hat{\sigma}_M^2, \hat{\sigma}_Y^2\right]\right) + \nu \Delta. \tag{21}$$

For example, this allows the evaluation of sensitivity with varying confounding strengths across mediator-outcome pairs. It can also be used to assess the effect of correlation across mediators. Note that when using either equations (20) and (21), we can simulate repeated data sets with the assumed covariance structure and refit models to estimate effects on each simulated data set. This allows us to report the standard error

of the estimated effects across choices of sensitivity analysis hyperparameters, helping to ensure that the sensitivity analysis itself is reliable.

## Microbiome-metabolome data processing

We obtained the data from the microbiome-metagenome curated database. Details of the library preparation and bioinformatics can be found in reference (58). Briefly, metagenomic sequencing was done on an Illumina HiSeq 2500, and metabolites were profiled using LC-MS in non-targeted mode. For metagenomics, fastp was applied to raw reads for quality filtering, adapter trimming, and deduplication. bowtie2 was used to remove human reads by aligning to the hg38. kraken2.1.1 and bracken 2.8 were used to estimate taxonomic relative abundances.

A total of 11,720 taxa and 8,848 metabolites are present in the public data. We applied a centered log-ratio transformation to the microbiome relative abundances profiles: $\mathrm{CLR}(x_1, ..., x_D) := \left(\log x_d - \frac{1}{D}\sum_{d'}\log x_{d'}\right)_{d=1}^{D}$. We then filtered to taxa whose average transformed abundance was larger than 3, which reduced the number of taxa to 173. We kept only metabolites with confident HMDB assignments, applied a $\log(1 + x)$ transformation, and further filtered to those whose average transformed intensity was larger than 6. This resulted in 155 well-annotated and generally abundant metabolites.

## Mindfulness study design and processing

The initial Center for Healthy Minds study recruited 114 police officers participants across two cohorts. Microbiome samples were obtained only from participants in the second cohort ($n = 54$), who were randomly assigned to mindfulness training or waitlist control with 27 cases each. We removed four participants due to incomplete responses—three lacked microbiome data, and one had missing mediators. Our analysis considers a mindfulness training treatment group of size $n = 24$ and a waitlist control group of size $n = 26$. Participants in the mindfulness group took part in an 8 week, 18 h mindfulness training developed specifically for their career and inspired by Mindfulness-Based Stress Reduction and Mindfulness-Based Resilience Training (47). Weekly 2-h classes (and a 4-h class in week 7) consisted of didactic instruction, embodied mindfulness practices, and individual and group-based inquiry [for full intervention details, see reference (48)]. Microbiota and behavioral survey data were gathered at 2–3 timepoints for each participant—samples in the treatment group provided data before, within 2 weeks following, and, in a subset of cases, 4 months after the 8 week intervention, resulting in 118 samples total.

Gut microbiome composition was assessed using 16S rRNA gene sequencing, and participants completed surveys, as reported previously (48). One to four technical replicates (on average, 2.6) were sequenced for each 16S rRNA gene sample, resulting in 307 microbiome composition profiles in total. Amplicon Sequence Variants (ASV) were called using the DADA2 pipeline (59). The first 10 base pairs were removed, and all reads were truncated to a length of 250. Otherwise, we set all pipeline hyperparameters to their defaults. Since the total number of participants is relatively small, we chose to concentrate on the core microbiome (60). To this end, we assigned taxonomic identity to each ASV using the RDP database and aggregated all counts to the genus level (61). We removed any genera that did not appear in at least 40% of the samples, thereby generating a core microbiome. On average, this preserved 98.7% of the reads within each sample. After filtering to the core microbiome, sequences for 55 genera remained. To define mediators, we manually selected four variables from the National Cancer Institute Quick Food Scan and self-reported questionnaires on fatigue and sleep disturbance scores based on the Patient-Reported Outcomes Measurement Information System subscale (62). We concentrated on these questions because changes in both diet and sleep have previously been associated with mindfulness interventions and the microbiome (63–65).

In detail, we consider four mediators—two diet mediators from the National Cancer Institute Quick Food Scan and two stress variables from the Patient-Reported Outcomes Measurement Information System (43-item inventory; version 2.0) following (62). They are all calculated from questionnaires. The two diet variables indicate the frequency that participants eat cold cereal and fruit (not juices), respectively, in the past 12 months (Table S1). The two stress variables, fatigue and sleep disturbance, profile the stress of a participant in the past 7 days (Table S2).

## ACKNOWLEDGMENTS

K.S. and J.H. were funded by award R01GM152744 from the National Institute of General Medical Sciences of the National Institutes of Health.

## AUTHOR AFFILIATIONS

[1]Statistics Department, University of Wisconsin—Madison, Madison, Wisconsin, USA
[2]Wisconsin Institute for Discovery, University of Wisconsin—Madison, Madison, Wisconsin, USA
[3]Center for Healthy Minds, University of Wisconsin—Madison, Madison, Wisconsin, USA
[4]Psychology Department, University of Wisconsin—Madison, Madison, Wisconsin, USA
[5]Psychiatry Department, University of Wisconsin—Madison, Madison, Wisconsin, USA
[6]Plant Pathology Department, University of Wisconsin—Madison, Madison, Wisconsin, USA

## AUTHOR ORCIDs

Margaret W. Thairu  https://orcid.org/0000-0002-2799-6261
Jo Handelsman  http://orcid.org/0000-0003-3488-5030
Kris Sankaran  http://orcid.org/0000-0002-9415-1971

## FUNDING

| Funder | Grant(s) | Author(s) |
|---|---|---|
| HHS \| NIH \| National Institute of General Medical Sciences (NIGMS) | R01GM152744 | Hanying Jiang |
| | | Jo Handelsman |
| | | Kris Sankaran |
| HHS \| National Institutes of Health (NIH) | T15 LM007359 | Margaret W. Thairu |
| | | Jo Handelsman |

## DATA AVAILABILITY

The multimedia package is available at https://go.wisc.edu/830110. Notebooks to reproduce the case studies are available at https://go.wisc.edu/787g25. These notebooks link to the original versions of both case study datasetsdata sets and include all preprocessing code. The package manual can be read at https://go.wisc.edu/olm213.

## ADDITIONAL FILES

The following material is available online.

### Supplemental Material

**Supplemental figures and tables (Spectrum01131-24-s0001.pdf).** Fig S1 and S2; Tables S1 and S2.

## Open Peer Review

**PEER REVIEW HISTORY (review-history.pdf).** An accounting of the reviewer comments and feedback.

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

Methods Protoc 8:bpad023. https://doi.org/10.1093/biomethods/bpad023

53. Chang JC, Allaire JJ, Sievert C, Schloerke B, YihuiXie JA, McPherson J, Dipert A, Borges B. 2024. Shiny: web application framework for R, 2024. R package ver 1919000. https://github.com/rstudio/shiny.

54. Imai K, Keele L, Yamamoto T. 2010. Identification, inference and sensitivity analysis for causal mediation effects. Statist Sci 25:51–71. https://doi.org/10.1214/10-STS321

55. Imai K, Yamamoto T. 2013. Identification and sensitivity analysis for multiple causal mechanisms: Revisiting evidence from framing experiments. Polit anal 21:141–171. https://doi.org/10.1093/pan/mps040

56. Friedman,J, Hastie,T, Tibshirani,R, Narasimhan B. 2023. Glmnet: Lasso and elastic-net regularized generalized linear models. Astrophysics Source Code Library, record ascl:2308.011, August 2023

57. Bürkner P-C. 2017. Advanced Bayesian Multilevel Modeling with the R Package brms. The R Journal 10:395. https://doi.org/10.32614/RJ-2018-017

58. Pasolli E, Schiffer L, Manghi P, Renson A, Obenchain V, Truong DT, Beghini F, Malik F, Ramos M, Dowd JB, Huttenhower C, Morgan M, Segata N, Waldron L. 2017. Accessible, curated metagenomic data through experimenthub. Nat Methods 14:1023–1024. https://doi.org/10.1038/nmeth.4468

59. Callahan BJ, McMurdie PJ, Rosen MJ, Han AW, Johnson AJA, Holmes SP. 2016. Dada2: high-resolution sample inference from illumina amplicon data. Nat Methods 13:581–583. https://doi.org/10.1038/nmeth.3869

60. Shade A, Handelsman J. 2012. Beyond the venn diagram: the hunt for a core microbiome. Environ Microbiol 14:4–12. https://doi.org/10.1111/j.1462-2920.2011.02585.x

61. Cole JR, Wang Q, Fish JA, Chai B, McGarrell DM, Sun Y, Brown CT, Porras-Alfaro A, Kuske CR, Tiedje JM. 2014. Ribosomal database project: data and tools for high throughput rRNA analysis. Nucleic Acids Res 42:D633–D642. https://doi.org/10.1093/nar/gkt1244

62. Cella D, Riley W, Stone A, Rothrock N, Reeve B, Yount S, Amtmann D, Bode R, Buysse D, Choi S, Cook K, Devellis R, DeWalt D, Fries JF, Gershon R, Hahn EA, Lai J-S, Pilkonis P, Revicki D, Rose M, Weinfurt K, Hays R, PROMIS Cooperative Group. 2010. The patient-reported outcomes measurement information system (PROMIS) developed and tested its first wave of adult self-reported health outcome item banks: 2005-2008. J Clin Epidemiol 63:1179–1194. https://doi.org/10.1016/j.jclinepi.2010.04.011

63. David LA, Maurice CF, Carmody RN, Gootenberg DB, Button JE, Wolfe BE, Ling AV, Devlin AS. 2014. Diet rapidly and reproducibly alters the human gut microbiome. Nat New Biol 505:559–563. https://doi.org/10.1038/nature12820

64. Waltz J, GilbertD. 2010 Mindfulness and health behaviors. Mindfulness (N Y) 1:227–234. https://doi.org/10.1007/s12671-010-0032-3

65. Wagner-Skacel J, Dalkner N, Moerkl S, Kreuzer K, Farzi A, Lackner S, Painold A, Reininghaus EZ, Butler MI, Bengesser S. 2020. Sleep and microbiome in psychiatric diseases. Nutrients 12:2198. https://doi.org/10.3390/nu12082198

