## [Reviewer comments · Microbiology Spectrum]

Microbiology Spectrum

multimedia: Multimodal Mediation Analysis of Microbiome Data

Hanying Jiang, Xinran Miao, Margaret Thairu, Mara Beebe, Dan Grupe, Richard Davidson, Jo Handelsman, and Kris Sankaran

Corresponding Author(s): Kris Sankaran, University of Wisconsin-Madison

Review Timeline:

Submission Date:	October 4, 2024
Editorial Decision:	October 21, 2024
Revision Received:	October 29, 2024
Accepted:	October 30, 2024

Editor: Jan Claesen

Reviewer(s): The reviewers have opted to remain anonymous.

Transaction Report:

DOI: <https://doi.org/10.1128/spectrum.01131-24>

Re: Spectrum01131-24 (multimedia: Multimodal Mediation Analysis of Microbiome Data)

Dear Prof. Kris Sankaran:

Thank you for the privilege of reviewing your work. Below you will find my comments, instructions from the Spectrum editorial office, and the reviewer comments.

Thank you for carefully addressing both Reviewers' comments. I feel these changes have strengthened your manuscript and I am pleased to inform you that your manuscript has been editorially accepted for publication. However, there are a few additional questions in the submission form that need to be answered before the final decision. Once these are completed, please return your submission so that I can move your paper forward to acceptance.

Revision Guidelines

Sincerely,
Jan Claesen
Editor
Microbiology Spectrum

Reviewer	Comment	Actions Taken
1	It should be clearly stated what this paper contributes, i.e., what are novel aspects of the proposed R package?	We now conclude the introduction with a bullet list summary of the novel contributions of our R package. Further, we have condensed the parts of the abstract that were focused on motivation and have extended the discussion of novelty ("enabling experimentation with ...making it well-suited to microbiome data.").
1	How are the direct and indirect effects estimated, or what are their estimators? Equations 7-10 are just general equations, whose results depend on types of outcome variables.	Our revised "Counterfactual framework" section now more clearly distinguishes between causal effect estimands (equations 7 - 9) and estimators (equations 15 - 17). Though equations 15 - 17 are still presented in their general form, we have added a new section, "Mediation and outcome model definition," that provides detail on how these estimators are implemented in our software via the generalized causal mediation analysis framework. Specifically, we explain the types of models that are available for estimating \hat{M} and \hat{Y} in those equations and we give the optimization objective functions that are used in the case studies. An important aspect of the package is that it allows general mediation and outcome model functions to be added after the fact by package users, as long as the functions conform to an S4 class definition that we provide. This point is now emphasized.
1	It seems that the CLR transformation is used to account for the compositional nature of microbiome data. Did the authors assume that this transformation makes mediators independent of each other? If not, how could indirect pathwise effects be estimated in the counterfactual framework?	We use the CLR transformation in Case Study 2 ("Microbiome-metabolome data integration"). However, the package can be used without necessarily applying a CLR transformation (e.g., as in Case Study 1). For Case Study 2, we do not assume that the mediators are independent, but we do require a form of sequential ignorability. We realize now that we did not provide sufficient discussion of the identification assumptions necessary for appropriate application of mediation analysis in microbiome data, and we have extended the discussion in the "Counterfactual framework" section of the Materials and Methods to clarify the identification assumptions, including alternative forms of sequential ignorability, needed in generalized causal mediation analysis with multiple mediators. References to the works that prove nonparametric identification for generalized causal mediation analysis with multiple mediators are also now provided.
1	For regularized regression, estimated coefficients are biased. How were these biases corrected to make valid inferences on indirect effects?	These effects are biased, and we did not correct them in our applications. This point is now clearly noted when we discuss the direct and indirect effects for the "Microbiome-metabolome data integration" case study where we apply regularized regression. We believe that debiasing mediation analysis estimates in high-dimensions is an active area of research, and it is a problem faced by a variety of high-dimensional, regularized mediation analysis models. Further, due to its modular design, our package is not tied to any individual model type, and we plan to continue developing templates using modern mediation analysis methods in future versions of the software.

1	Random forests is a prediction model. Could the authors explain how their estimates could be used to estimate and test indirect effects?	This is related to the Question 3 above. We believe that the new section "Medation and outcome model definition" clarifies how we are able adapt prediction methods to support direct and indirect effect estimation within the generalized causal mediation analysis framework. Given those model-based estimates (e.g., from random forests, logistic-normal multinomial, regularized regression, or hierarchical models), we form confidence intervals/FDR-controlling significance tests using the approach in Section "Bootstrap and synthetic null testing." We have also added this summary of the high-level philosophy for the package, which gets at exactly your question: "Therefore, advances in prediction of microbiome data can be easily incorporated to improve causal effect estimation through higher-quality mediator and outcome models."
1	In mediation analysis, it is extremely important to reduce potential confounding effects and assess their sensitivity. Does the proposed R package provide ways to reduce the confounding effects and do sensitivity analysis?	We agree that sensitivity analysis is essential for the interpretation of indirect effects. In response to this comment, we have introduced three new functions to the package for evaluating sensitivity analysis, listed in the updated Table 1. These functions provide simulation-based analogs to Theorem 4 in Imai, Keele, and Yamamoto (2010). The approach is detailed in a new section, "Sensitivity analysis," in the Materials and Methods. We have also added a discussion of sensitivity in the "Microbiome-metabolome data integration" case study, including the new Figure 6. One limitation of our approach is that, like Imai, Keele, and Yamamoto (2010), it relies on introducing correlation in additive noise terms of the mediator and outcome models. Therefore, it is not directly applicable to models with non-additive noise structure, including the logistic-normal multinomial and hurdle models that we provide. We make this point at the start at the start of the sensitivity analysis section, so that potential users clearly understand the capabilities and limitations of our software.
2	The release of the R package "multimedia" on CRAN instead of Github would be preferable in order to provide a more stable application environment.	We appreciate this comment and have revised the package and documentation to ensure long-term stability. The package is also now hosted on CRAN (https://cran.r-project.org/package=multimedia), and we have written a suite of tests with 90% code coverage (https://app.codecov.io/github/krisrs1128/multimedia). These tests are executed each time an update is pushed to the package repository, so we will learn early on whenever a breaking change appears. We have also created a mybinder.org notebook for potential users to experiment with before installing the package locally. We will use code coverage, CRAN status messages, and binder notebook status to ensure long-term viability of our software.
2	Could you kindly provide a manual for the "multimedia", similar to those available for R packages on CRAN?	Now that the package is hosted on CRAN, this manual can be accessed at the CRAN package page. We have also provided a direct link (https://go.wisc.edu/olm213) in both the table describing package functions and the data availability statement.

2 Are the datasets of two case studies publicly available in the "multimedia"? or could you please provide more details on how to access these datasets mentioned in the paper?		Yes, the datasets are available! The package's vignettes are designed to be reproducible, and the code for loading the data are included within them. For example, the line <pre>data(mindfulness)</pre> in the CHM study vignette (https://krisrs1128.github.io/multimedia/articles/mindfulness.html) and the lines <pre>Sys.setenv("VROOM_CONNECTION_SIZE" = 5e6) taxa <- read_tsv("https://go.wisc.edu/I015v0"), -1] metabolites <- read_tsv("https://go.wisc.edu/0t3gs3"), -1] metadata <- read_tsv("https://go.wisc.edu/9z36wr")</pre> in the IBD microbiomics-metabolomics case study read in the data for those two studies, respectively. The code there also transforms the raw data into preprocessed phyloseq objects. We have revised the data availability statement to highlight the fact that these data are available. In particular, we have added: "These notebooks link to the original versions of both case study datasets and include all preprocessing code."
2 Jang et al. (2023) has also provided a web cloud computing platform, named as MiMed, for microbiome causal mediation analysis. The difference and advantage of the R package "multimedia" compared to "MiMed" should be discussed.		Thank you for bringing this important related work to our attention. We have added a new paragraph to the discussion to compare and contrast the two packages. Overall, we believe that they serve complementary roles in the software ecosystem for microbiome mediation analysis, and we explain this in detail within our discussion.
2 The microbiome data (OTU) is characterized by high dimensionality, zero-inflation, and compositional nature (Sohn and Li, 2019; Wang et al. 2020). However, the paper does not address or tackle these challenges adequately. Could you please provide more details on this issue?		We realize that we may not have communicated this clearly in our original submission. The intent of the package is to make microbiome mediation analysis adaptable to a variety of mediator and outcome model forms. Rather than seeking a one-size-fits-all solution to the important issues that you have raised, we seek to make it easier to compare, diagnose, and modify models so that they adequately address these issues. The specific mediator and outcome model components that are provided by default aid in this search, because they are well-suited to address these issues. For example, glmnet is well-suited to high-dimensional data, the LNM is designed for compositional data, and the hurdle-log normal model is designed for zero-inflation structures. We now elaborate further on this point in the abstract, the bullet list of contributions, and the "Mediation models definition" section of the Materials and Methods.

Re: Spectrum01131-24R1 (multimedia: Multimodal Mediation Analysis of Microbiome Data)

Dear Prof. Kris Sankaran:

Thanks for addressing the additional Spectrum-specific submission questions. I hereby would like to congratulate you on the acceptance of your manuscript for publication! Thanks for making your tool available to the community, I think this will be valuable to many and am excited to try it out as well.

Your manuscript has been accepted, and I am forwarding it to the ASM production staff for publication. Your paper will first be checked to make sure all elements meet the technical requirements. ASM staff will contact you if anything needs to be revised before copyediting and production can begin. Otherwise, you will be notified when your proofs are ready to be viewed.

Sincerely,
Jan Claesen
Editor
Microbiology Spectrum